# Exploiting Tomato Genotypes to Understand Heat Stress Tolerance

**DOI:** 10.3390/plants11223170

**Published:** 2022-11-19

**Authors:** Emma Fernández-Crespo, Luisa Liu-Xu, Carlos Albert-Sidro, Loredana Scalschi, Eugenio Llorens, Ana Isabel González-Hernández, Oscar Crespo, Carmen Gonzalez-Bosch, Gemma Camañes, Pilar García-Agustín, Begonya Vicedo

**Affiliations:** 1Grupo de Bioquímica y Biotecnología, Área de Fisiología Vegetal, Departamento de Biología, Bioquímica y Ciencias Naturales, ESTCE, Universitat Jaume I, 12071 Castellón, Spain; 2Departament de Bioquímica, Instituto de Agroquímica y Tecnología de Alimentos (CSIC), Universitat de València, 46980 Valencia, Spain

**Keywords:** heat stress, tomato, thermotolerance

## Abstract

Increased temperatures caused by climate change constitute a significant threat to agriculture and food security. The selection of improved crop varieties with greater tolerance to heat stress is crucial for the future of agriculture. To overcome this challenge, four traditional tomato varieties from the Mediterranean basin and two commercial genotypes were selected to characterize their responses at high temperatures. The screening of phenotypes under heat shock conditions allowed to classify the tomato genotypes as: heat-sensitive: TH-30, ADX2; intermediate: ISR-10 and Ailsa Craig; heat-tolerant: MM and MO-10. These results reveal the intra-genetical variation of heat stress responses, which can be exploited as promising sources of tolerance to climate change conditions. Two different thermotolerance strategies were observed. The MO-10 plants tolerance was based on the control of the leaf cooling mechanism and the rapid *RBOHB* activation and ABA signaling pathways. The variety MM displayed a different strategy based on the activation of *HSP70* and *90*, as well as accumulation of phenolic compounds correlated with early induction of *PAL* expression. The importance of secondary metabolism in the recovery phase has been also revealed. Understanding the molecular events allowing plants to overcome heat stress constitutes a promising approach for selecting climate resilient tomato varieties.

## 1. Introduction

Heat stress (HS) is one of the most important abiotic stresses that limit crop productivity worldwide [1]. In the current scenario of global warming, it is expected that temperatures will rise between 2 and 5 °C by the end of the 21st century [2,3]. The average of global surface temperature, will cause serious damage on growth and development of plants, resulting in a catastrophic reduction of crop productivity [1,4]. 

In addition to the progressive increase in temperatures, another threat to crop productivity is the increased frequency of extreme weather events such as heat waves, floods or prolonged droughts [5]. Therefore, comprehending the response mechanisms of plants to abiotic stress (specially heat stress), the adaptation of seed varieties and the management of crops at high temperatures will be key to the agronomic sustenance [6]. In addition, the increased production required to meet the nutritional needs of a constantly growing world population (9 billion by the year 2050) together with the expected crop losses, is a challenge that can only be solved by agricultural production systems based on the use of crop varieties that are more tolerant to abiotic stress. 

The negative effects of increased temperatures on plant growth and productivity have been already described in crops such as wheat, rice, barley, sorghum or maize have al-ready been described [4,7,8]. Although all plant tissues are severely affected by increased temperatures, the reproductive tissues are the most sensitive, and a temperature rise of a few degrees during the flowering period can cause complete loss of growth harvest as occurs in grain crops [7]. Tomato is considered particularly sensitive to heat stress since, the increase in temperature produces yield losses of up to 28% [9]. The increase of 1 °C up to its optimal growth temperature can greatly impair pollen viability and female fertility, seriously affecting fruit set [10]. Moreover, it is commonly accepted that the reduction of the life cycle that decreases plant productivity upon heat stress is produced by the impact of this stress on respiration and photosynthesis rates [11]. 

Plants as sessile organisms have evolved a series of strategies to ensure survival and reproductive success when confronting heat stress. The ability of plants to survive abrupt temperature increases is known as basal thermotolerance [12]. There are some strategies in charge of inducing acclimation mechanisms or to avoid the stress produced by the in-crease in short-term temperatures, such as the reorientation of the different organs, the manifestations of the lipid composition of the membrane or the increase of transpiration rate to allow leaf cooling [13]. It is commonly accepted that transpiration cooling is an important process in thermotolerance. This process is based on keeping the stomata open during high temperature stress, which allows the diffusion of CO_2_ through the leaf blade, which allows plants to control leaf temperature [14,15]. A recent study demonstrated that heat tolerant common bean genotypes cool more than heat sensitive genotypes as a result of higher stomatal conductance and enhanced transpiration cooling [16].

Plants under the increase of temperature conditions will activate and accumulate heat shock proteins (HSPs), including HSP100, HSP90, HSP70, HSP60, and small HSPs (smHSPs) [17]. These HSPs are considered crucial molecular chaperones that participate in the thermotolerance responses to maintain the stability of cells [18]. In addition, heat shock transcription factors (HSFs) are responsible for regulation of HSPs, which control protein quality [19]. For example, the overexpression of heat stress transcription factor A2 of Oryza sativa in Arabidopsis induced the upregulation of HSPs what enhance thermotolerance [20,21]. 

As occurs in other stress situations, heat stress produces secondary stress in plants such as oxidative stress, caused by damage to chlorophylls and the photosynthetic apparatus, which results in an overproduction of reactive oxygen species (ROS) [22]. Heat stress affects the membrane fluidity, producing increased cytosolic calcium that transduced via respiratory burst oxidase homolog (RBOH) proteins, initiating the ROS burst at the apoplast [23,24]. Thus, the increase of both signaling molecules contributes to activate downstream signaling pathways that regulates HSFs, initiating therefore, heat stress responses [25]. Plants also induced the expression of antioxidant proteins like ascorbate peroxidase (APX), catalase (CAT), or superoxide dismutase (SOD) to counteract the negative effects of excessive ROS under high temperatures [26]. The importance of ROS-scavenging enzymes as well as other antioxidant molecules like ascorbic acid in thermotolerance has been largely demonstrated by using knockout mutants what was recently reviewed by [25]. 

Plant hormones, including abscisic acid (ABA), brassinosteroids (BRs), cytokinins (CKs), salicylic acid (SA), jasmonic acid (JA), and ethylene (ET), play important roles con-trolling complex stress-adaptive signaling cascades that triggered heat stress responses [27,28]. The role of ABA and SA in basal thermotolerance has been extensively studied. Exogenous application of ABA 5 µM on tall fescue produced an enhancement of heat tolerance based on the increase of leaf photochemical efficiency and membrane stability [29]. Likewise, Arabidopsis microarray suggests that ABA can induce thermotolerance by inducing the expression of HSFA6b [30]. Previous studies suggest that SA is primarily involved in promoting the basal thermotolerance by inducing several HSPs [31]. However, recent studies showed the relevance of this phytohormone in the thermotolerance mechanisms by regulating the antioxidant defense system and improving photosynthetic efficiency upon heat stress [32,33].

In addition to the classical ABA- and SA-mediated responses, tomato plants respond to environmental changes like temperature increase producing significant changes in the phenolics and flavonoids contents [34,35]. These secondary metabolites play a key role in the protection of plants against unfavorable situations "through their antioxidant capacity since they’re able to inhibit ROS formation via a range of different mechanisms [36]. Plant phenolics are synthesized through the shikimate/phenylpropanoid pathway from phenylalanine. Tryptophan is another relevant amino acid because is precursor of the hormone melatonin which is involved in plant growth and development and plays a key role in a wide range of abiotic stresses [37]. In the recent years, the importance of this hormone in the plant responses against heat stress is becoming clear, since it is involved in the improvement of photosynthetic efficiency, the regulation of stomatal movement and the synthesis of chlorophyll and RuBisCo activity [38,39,40]. Thermotolerance is improved by treating tomato and wheat plants with melatonin, which also maintains membrane stability, plant water relations, and increases antioxidant activities [41,42,43]. 

A general response mechanism of plants against heat stress has been stated, but it is well known that these defenses vary according to the developmental stage, genotype, species and heat stress experimental conditions [44,45,46]. In fact, resistance to heat shock (HS) is genetically diverse [47,48] and therefore it is important to exploit this genetic diversity to elucidate the mechanisms that allow certain genotypes to grow and have greater yield under warm temperatures, being of great interest for breeding programs to select tolerant varieties to heat [49]. Landraces, or traditional varieties defined by [50] as “a dynamic population of a cultivated plant that has historical origin, distinct identity and lacks formal crop improvement, as well as often being genetically diverse, locally adapted and associated with traditional farming systems”, are one of the most important components of plant genetic resource. Nowadays, tomato landraces are still cultivated for local use and consumption [51] The adaptability of landraces to unfavorable environmental situations [52] translates into the possibility of finding new genes or strategies for thermo-tolerance, which can be used in breeding and selecting climate-resilient plants. In addition to the traits that produce thermotolerance in plants, identifying the mechanisms that help to overcome the damages produced by heats shocks or waves once the stress is release, is a key point for plant breeding in the fight against climate change. However, these recovery processes are actually poorly understood [53]. 

The main objective of this work is the characterization of traditional tomato varieties (or tomato landraces) of different areas of the Mediterranean basin against heat stress. To accomplish this purpose, four traditional varieties *S. lycopersicum* genotypes: TH-30 (Greece); ADX2 (Spain); ISR-10 (Israel); MO-10 (France) and two commercial genotypes: Ailsa Craig (Ailsa) and Moneymaker (MM), were screened for heat tolerance. To understand the intra-genetical variation, the main defense signaling pathways activated to counteract the negative effect of increased temperature were evaluated. We provide evidence of the relevance of leaf cooling mechanisms and the regulation of the oxidative burst produced by HS. We also demonstrated that SA, phenolic compounds, and melatonin are crucial components for overcoming HS. Understanding the molecular events that take place in the different phenotypes will help us get a deeper understanding of plant response to heat stress, which can be fundamental for the maintenance of food production in the near future.

## 2. Results 

### 2.1. Screening Tomato Phenotypical Variations in Thermotolerance 

To achieve HS, the temperature of the growth chamber was increased to 42 °C for 6 h. Plant recovery capacity was subsequently evaluated after two hours under normal temperature conditions. In order to check how HS could affect different tomato genotypes, leaf damage was evaluated at 2, 4, 6 and 6+2 hpHS (hours post-heat shock) and the results were expressed as the percentage of damaged leaflets of third and fourth true leaves. Leaf damage was measured using a four-level severity scale that determined the damage depending on the rolling of the leaflets: level 0 (healthy), level 1 (minor roll), level 2 (medium roll) and level 3 (severe roll or dead). At 2 hpHS, we observed that TH-30, Ailsa and ADX2 genotypes displayed higher leaf damage than the other tomato genotypes. Leaf damage was especially important in ADX2 plants, which had more than 10% of their leaflets in level 2 (Figure 1a). ISR-10, MM, and MO-10 genotypes showed less than 50% damaged leaflets (most of them in level 1) after 4 hpHS. Moreover, at this time point a significant damage on TH-30, Ailsa, and ADX2 leaves was observed, since they exhibited 80, 70, and 50% of damaged leaflets, respectively (Figure 1b). At the climax of stress (6hpHS), clear phenotypic differences were observed among the studied tomato genotypes. TH-30 plants were strongly affected by the increase of temperature, showing 100% of its leaflets damaged (50% in level 3). Ailsa and ADX2 genotypes showed more than 70% damaged leaflets. On the opposite, we found that MM and MO-10 genotypes displayed less than 60% damaged leaflets (only 20% in levels 2 and 3) showing a tolerant phenotype (Figure 1c). 

Once classified the tomato genotypes depending on their thermotolerance, we evaluated their behavior in the recovery phase. For this purpose, the visual phenotype was evaluated after 2 h of absence of stress without water added.

Figure 1d shows leaflet damage after 6 h of heat shock and 2 h of recovery (6+2 hpHS). ISR-10 plants showed 100% of leaflets in level 0 (full recovery). Moreover, Ailsa and ADX2 displayed the 80% of their leaflets in level 0, an almost full recovery. However, TH-30 plants still revealed to have 50% damaged leaflets at this time point (part of them in levels 2 and 3). MM and MO-10 plants displayed 30% damaged leaflets although most of them were in level 3. On the other hand, the plants of MO-10 genotype displayed 50% damaged leaflets but most of them were found at level 1.

### 2.2. Effect of Time of HS on the Relative Water Content (RWC)

To characterize the response of the different tomato genotypes to high temperature stress, the relative water content (RWC) of each leaflet was evaluated, since it is known that high temperature results in rapid loss of water that may cause dehydration [54]. We observed a HS related RWC reduction on all tomato genotypes, though significant differences between them were detected (Figure 2). At 2 hpHS, it was observed that TH-30, ADX2 and ISR-10 genotypes displayed a clear reduction in RWC that reached as low as 75%. However, MM plants were able to maintain high RWC levels after 2 h at 42 °C. At 4 hpHS, plants of MM and MO-10 genotypes displayed a slight reduction of RWC content. However, ISR-10, Ailsa, and TH-30 plants showed reduction of 11.8, 12.8, and 11.1%, respectively, comparing with their basal values indicating that the differences in dehydration produced by the increase of external temperatures could be correlated with the different sensitivities of the varieties to HS. The most significant changes associated with the reduction of RWC on tomato leaves was observed at 6 hpHS. MM and MO-10 plants maintained the RWC of their leaves at a similar level to basal ones throughout the HS process. For Ailsa and ISR-10, although they displayed an initial loss of water in the first two hours of HS, RWC was unaffected at 4 and 6 hpHS maintaining a value of 73%. Finally, plants of ADX2 and TH-30 genotypes showed at 6hpHS a noticeable reduction in RWC content induced by the HS, which is especially important in the TH-30 genotype. In the recovery phase, (6+2 hpH), plants of MM, MO-10, and ISR-10 varieties, which showed less water loss during HS, maintained similar RWC after two hours in the absence of stress. However, genotypes such as Ailsa, TH-30, and ADX2 that displayed a severe decline during the HS process, recovered to values around 77–80%.

### 2.3. Effect of HS on Photosynthetic Parameters of Different Tomato Genotypes

In order to determine the tolerance or susceptibility of tomato genotypes to HS, different photosynthetic parameters were analyzed (Figure 3). Interestingly, in the absence of HS, the genotypes showed statistically significative differences regarding photosynthetic parameters (Figure 3a). As expected, all tomato genotypes showed a significant reduction of A through HS; however, TH-30 and ADX2 plants displayed a strong reduction of A at 2, 4, and 6 hpHS. Surprisingly, the MO-10 plants showed a slight reduction of A at 2 and 4 hpHS compared with its basal levels, but it showed a significative A decrease at 6 hpHS MM plants displayed a strongly reduced A at 2 hpHS, showing a recovery at 4 hpHS and another marked reduction at 6 hpHS, reaching values of 6 μmol CO_2_ m^−2^ s^−1^. Although at 6+2 hpHS, in the recovery phase, all genotypes displayed lower A when compared to their basal values, statistical differences were seen between the genotypes. MM, MO-10, ISR-10, and Ailsa that showed photosynthetic rates close to 6, while the most affected genotypes, TH-30 and ADX2, presented a rate of 0.27 and 0.44 μmol CO_2_ m^−2^ s^−1^, respectively. 

The transpiration rate (E) and stomatal conductance (g_s_) of the tomato genotypes selected for the study are represented at Figure 3b,c. The most significant changes on E and g_s_ occur at 2 hpHS, where all genotypes displayed a decrease of both parameters being this reduction especially important in MM, Ailsa, and TH-30 plants. ISR-10 and ADX2 genotypes showed a strong reduction when compared to their basal values; however, these plants maintained higher levels of both parameters than the susceptible genotypes. In the late stage of HS, significant differences were seen in E and g_s_, where TH-30 and ADX2 plants showed the lowest values of both parameters, demonstrating the HS damage. In addition, ISR-10, MM, and MO-10 genotypes revealed a strong reduction of E and g_s_ at 6 hpHS compared to their basal values but presented higher values than the most affected genotypes without statistically significant differences among them. Interestingly, no variation in the E and g_s_ values were found in the recovery phase between genotypes. 

On the other hand, leaf temperature (Tleaf, Figure 3d) was evaluated throughout the HS in the six genotypes under study. At 2 hpHS, an increase in the Tleaf in all groups of plants was observed. Still, this increase was especially significant in TH-30, MM, and ADX2 plants. Moreover, at 4 hpHS, Tleaf was the highest temperature scored during the HS, reaching values of 28.39, 28.42, and 28.28 °C in the TH-30, ADX2, and ISR-10 genotypes, respectively. At 6 hpHS all genotypes were able to reduce Tleaf value, MM plants being the ones that displayed the lowest foliar temperature (Figure 3d).

### 2.4. Analyzing the Changes in Fv′/Fm′ and ETR to Characterize Tomato Genotypes as a Heat-Sensitive or Heat-Tolerant

The maximum quantum efficiency of photosystem II (Fv′/Fm′) is used to quantify damage to Photosystem II. This parameter has been used to evaluate stress response in many plants’ species [48]. In this study, Fv′/Fm′ ratio has been analyzed to evaluate the thermotolerance of different tomato varieties. Healthy plants hold Fv′/Fm′ values that range around 0.8, which decrease upon stress. In our experimental procedure, basal Fv′/Fm′ ranged around 0.75+SE. After 2 hpHS, an important reduction of Fv′/Fm′ ratio was observed in ADX2, Ailsa, MM, and TH-30 plants compared to their respective basal values, while MO-10 and ISR-10 plants were able to maintain similar basal ratio at this time point. At 6 and 6+2 hpHS, TH-30 and ADX2 genotypes displayed Fv′/Fm′ ratios between 0.4–0.5 demonstrating the serious leaf damage due to increased temperature (Figure 4a). 

In addition to Fv′/Fm′, another stress marker was evaluated. As expected, HS produced in tomato genotypes a reduction of electron transport rate (ETR). However, ETR reduction was statistically significant in TH-30 and ADX2 plants, with reductions of 71 and 49% at 6 hpHS relative to their basal values. On the other hand, MO-10 plants displayed high ETR values at 6 and 6+2 hpHS confirming lower level of cell damage by HS (Figure 4b).

Based on visual damage evaluation, leaf RWC, photosynthetic parameters and parameters related to photosystems damage, a scale of thermotolerance was established considering: thermo-sensitive genotypes TH-30 and ADX2; genotypes with intermediate behavior, ISR-10 and Ailsa; and thermo-tolerant genotypes, MM and MO-10. To gain further insight into the biochemical and molecular mechanisms related to thermotolerance in tomato plants, two susceptible (TH-30 and ADX2) and two tolerant (MM and MO-10) genotypes were selected to continue with the characterization. 

### 2.5. Modification of HSP Pattern Expression among Tomato Genotype under HS

To determine whether the tomato responses against HS are genotype-dependent, the expression of marker genes for thermotolerance-related pathways was evaluated (Figure 5). First, the expression of heat shock proteins was analyzed, specifically *HSP70* and *HSP90* and the transcription factor *HSFA2*, which is involved in their regulation. In all the studied genotypes, an early induction of *HSP70* was observed at 2 hpHS (Figure 5a). Moreover, TH-30 and MM plants had an exponential increase of *HSP70* transcript levels at 2 hpHS. Regarding to the expression of the *HSP90* gene (Figure 5b), a strong induction was observed at 2 hpHS in all genotypes, being this increase strongly marked on MM plants compared to MO-10 genotypes. Interestingly, the genotypes considered tolerants, MM and MO-10, showed a slight induction of this gene in the recovery phase. Figure 5c shows the expression values of *HSFA2*, which is induced in all the varieties at 2 and 6 hpSH, specially in TH-30 plants. 

### 2.6. Control of Antioxidant Machinery Is Crucial in Thermotolerance 

In addition to the study of chaperones for characterizing the response of different tomato genotypes to HS, genes related to the oxidative burst produced in plants after a stress situation and the antioxidant genes were analyzed (Figure 6). The obtained results of *RBOHB* expression (Figure 6a) showed that at early phase of HS, it was activated in MO-10 plants. At 6 hpHS, a strong induction of this gen was exhibited in both MO-10 and MM genotypes, which was maintained in the recovery phase. In general, genes related to oxidative stress were more induced in those genotypes that displayed greater tolerance phenotype against HS. Specifically, MO-10 plants showed an early activation of the *DHAR1* gene when compared to the rest of genotypes. In these plants, a statistically significant increase in *SODct* expression was also observed at 6 hpHS. Moreover, MM plants showed an induction of *CAT* gene at 6 hpHS, which remained in the recovery phase. Hence, control of antioxidant systems seems to have a clear role in tomato thermotolerance. 

### 2.7. Tolerant Genotypes Displayed an Early Activation of ABA-Dependent Signaling Pathways under HS

To establish the molecular mechanisms that underlie the thermotolerance observed in some of the studied genotypes, we analyzed the hormonal and metabolite content in tomato leaves under HS (Figure 7). Regarding ABA accumulation, although all genotypes displayed slight changes on ABA content during the HS, MO-10 plants showed statistically significant accumulation of ABA at 2 hpHS. On the other hand, TH-30 plants displayed a higher accumulation of ABA at 6 hpHS and these levels were increased during the recovery phase, reaching values of 850 ng/g FW (Figure 7a). To delve into the role of ABA in the tolerance or susceptibility of different tomato genotypes against HS, the expression of the *ASR1* gene has been evaluated (Figure 7b). The increase of temperature induced the transcript levels of *ASR1* in MM and MO-10 plants in an early phase of HS. Moreover, at 6 hpHS, although we found an induction of the *ASR1* gene in all group of plants against HS, the expression was higher in the plants of the tolerant genotypes MM and MO-10. Although HS did not increase the SA levels of any of the tomato genotypes (Figure 7c), a significant accumulation of SA occurs in the recovery phase in all group of plants, where TH-30 and ADX2 genotypes showed the highest levels of SA. 

### 2.8. Phenolic Compounds and Melatonin Synthesis Seem to Be Key for the Recovery of Cells after HS

To clarify the mechanisms involved in plant thermotolerance, the metabolic pathways related with the synthesis of active compounds against abiotic stresses were studied. First, the phenylpropanoid pathway was analyzed in four tomato genotypes at 2, 4, and 6 hpHS as well as in the recovery phase. The MM plants, cataloged as a tolerant genotype, displayed large increase of *PAL* transcript accumulation at 2 hpHS (Figure 8a). The activation of *PAL* correlated with an increase in chlorogenic acid (CGA) in these plants at 4 hpHS (Figure 8b). Interestingly, this phenolic compound was differentially accumulated at 6 hpHS in genotypes considered thermo-sensitive (TH-30 and ADX2). In the recovery phase, susceptible genotypes displayed a great induction of *PAL* expression, which produced an increase of CGA content in plants of TH-30 and ADX2 genotypes. This last genotype showed a significant increase of ferulic acid (FA) at 6+2 hpHS. 

Finally, the role of melatonin in the plant responses against the temperature rise was evaluated. To do this, we evaluated the expression of *ASMT*, which encodes a protein that catalyzes the last step of the melatonin biosynthetic pathway (Figure 8d). *ASMT* was induced by HS in all tomato genotypes at 2 hpHS. However, at 6 hpHS, TH-30 plants displayed a greater induction of *ASMT* gene compared to the other genotypes. Interestingly, the highest levels of *ASMT* expression were found in the recovery phase without statistically significant changes between the genotypes. Taken together, these results suggest the involvement of phenylpropanoid-derived compounds and melatonin in the recovery of tomato plants after a HS situation. 

## 3. Discussion

Increased temperatures caused by climate change are a significant threat to agriculture and food security. Current models predict an increase in the global average temperature between 2 and 5 °C over the next hundred years. Extreme weather events, which include heat waves, will have devastating consequences for crop productivity [1,4]. Along with heat stress, extreme drought is the other main abiotic stress directly related to climate change, and both will affect the Mediterranean basin more severely than other areas of the world. For this reason, the development of varieties with greater tolerance to these extreme weather events is considered a vital need for the future of agriculture [55]. 

To overcome this predicted situation, special attention is being paid to the genetic potential that wild and traditional plants constitute, since they represent a genetic reservoir not yet explored. In this study, we analyzed the response of different tomato genotypes to HS from different perspectives with the purpose of finding which set of responses are the key mechanisms to thermotolerance. To do so, we selected four traditional *S. lycopersicum* genotypes, originated from the Mediterranean area: TH-30 (Greece); ADX2 (Spain); ISR-10 (Israel); MO-10 (France) and two commercial genotypes: *Ailsa Craig* (Ailsa) and Moneymaker (MM).

HS was induced by increasing growth chamber temperature to 42 °C for 6 h, simulating the environmental conditions of an extreme summer. In addition, we analyzed the plant recovery 2 h post-stress what reflects plant thermotolerance. After checking for any damage induced by HS, we found a differential response to this unfavorable situation among genotypes. Specifically, the visual evaluation of foliar damage revealed that the most affected genotypes by the increase in temperature were TH-30 and ADX2, showing 100% and 70% of its leaflets damaged in level 3, respectively, and the most tolerant were MM and MO-10, since these genotypes showed less than 60% damaged leaflets observing only 20% of them in levels 2 and 3 respectively. We also evaluated RWC, which is considered an indicator of water condition of cells and correlated with tolerance or susceptibility to biotic and abiotic stresses [56]. Our results are in concordance with this idea since HS produced a reduction of RWC content in the six tomato genotypes. MM and MO-10 displayed a slight change in RWC content, whereas TH-30 and ADX2 plants suffered great dehydration during HS. 

Another common effect of heat stress on plants is the impact on photosynthesis, mainly through affecting biochemical reactions [57]. Although a general reduction in A was observed during HS, we found clear differences among genotypes. At 4 hpHS, TH-30 plants displayed a strong reduction of A rate, followed by the ADX2, Ailsa and ISR-10 genotypes, which displayed a strong reduction of A rate compared to their basal values. Nevertheless, MM, and MO-10 were able to maintain higher A during the HS stress. The ability to sustain leaf gas exchange under HS has been directly correlated with heat tolerance [1]. Recently, it has been demonstrated that heat tolerant wheat cultivars could maintain high rates of photosynthesis and stomatal conductance during HS [46]. 

The literature demonstrates that transpirational cooling is an important mechanism for heat avoidance in food crops [16]. In fact, the results obtained for transpiration and stomatal closure, Tleaf and cell damage parameters are probably the core point in the intra-genetical thermotolerance variation obtained in this work. As expected, plants of TH-30, Ailsa, and MM genotypes showed a sudden reduction in transpiration (E) and stomatal conductance (g_s_) values during heat stress. However, no changes of E and gs were observed in the early phase of HS in MO-10 plants when compared to the values obtained in stress absence. These results suggest that the capacity of MO-10 to maintain high levels of E and gs during HS might be the key event related to the tolerance phenotype since these processes contribute to leaf cooling, allowing to keep leaf temperature within the range required to maintain its optimal physiological function [16]. Therefore, the maintenance of the transpirational cooling in MO-10 plants might regulate the temperature of the leaves, keeping it almost 0.5 °C below the foliar T° observed in the susceptible genotypes. Moreover, in this work, we have studied the cell damage produced by heat stress (measured based on the reduction in both ETR and Fv′/Fm′ parameters) observing that MO-10 plants did not show the cell damage detected at 2 hpHS in the other genotypes. This demonstrates that the genotypes capable of activating the leaf blade cooling process by maintaining E and g_s_, are also able to reduce the excessive increase in Tleaf that could avoiding this way damages to the cellular structures. This hypothesis is confirmed by the results obtained in the MM genotype, since at 2 hpHS these plants displayed a decrease of E and g_s_, that probably reduces the leaf cooling. This T° increase could produce cell damage as demonstrated by the increment observed for both ETR and Fv′/Fm′ parameters. However, at 4 hpHS, MM plants opened stomata and increased the transpiration rate up to 0.0014 mol H_2_O m^−2^ s^−1^ XX (reaching values observed in MO-10 plants), that correlates with a reduction of Tleaf, which normalizes the Fv′/Fm′ and ETR values.

According to these findings, the mechanism of transpiration cooling is a key event in the response of tomato plants to HS since it allows the maintenance of leaf temperature within optimal ranges that ensure the correct cell function. After the analysis of the visual phenotype, the RWC and the evaluation of the photosynthetic parameters in different tomato genotypes under HS, we established a thermotolerance classification. The obtained results demonstrated that the genetical variation of tomato plants responses against HS exists, which allowed to characterize the genotypes studied in this work as: heat-sensitive: TH-30, ADX2; intermediate: ISR-10 and Ailsa; heat-tolerant: MM and MO-10.

To understand the mechanisms activated in tolerant tomato genotypes against HS, a comparative study was performed between heat-sensitive (TH-30 and ADX2) versus heat-tolerant (MM and MO-10) genotypes. 

The induction of *HSFs* and *HSPs* is considered essential for plant thermotolerance. As expected, all tomato genotypes displayed a strongly increase in the expression of *HSP70* and *90*, encoding chaperones related with plants response to heat stress, and the transcription factor *HSFA2* involved in their regulation. Despite the rapid and marked increase in these transcripts in all of the genotypes studied, surprising differences among them were detected. TH-30 plants showed the highest induction of *HSFA2* expression at 2 and 6 hpHS, that did not correlate with an increase of thermotolerance. Interestingly, the highest levels of *HSP70* expression were observed in TH-30 and MM genotypes, which displayed an opposite visual phenotype. The importance of *HSP70* in the plant responses against heat stress is noteworthy since its expression correlated with an increase of thermotolerance in *Arabidopsis* [58]. Interestingly, [59] reported that *Hsp70B* found in the stroma of chloroplasts participates in photo protection and the repairing of photosystem II during and after the photoinhibition. According to these findings, it can be speculated that a higher expression in *HSP70* is correlated with a severe damage on the photosystems in TH-30 and MM plants probably due to a poor control of the leaf cooling process that increased leaf temperature. Regarding *HSP90* expression, a strong induction was observed in all genotypes, but the highest level of induction was found in the MM plants. The crucial role of HSP40/HSP70 chaperone machinery in abiotic stress response is a well-known mechanism [60]. However, *HSP90* expression has been related to direct protection against abiotic stresses almost as a transduction signal [61]. For example, the expression of *Glycine max HSP90* in *Arabidopsis* protected plants against heat, salt, and oxidative stress [62]. The synergic effect of *HSP70* and *HSP90* activation in MM might have an important role in the tolerant phenotype showed by these plants against HS despite the initial damage produced by a sudden increase of temperature. It can be speculated that *HSP70* activation could be related with the rebuilding of photosynthetic apparatus and that *HSP90* induction could activate mechanisms related to thermotolerance.

To elucidate the mechanisms related to stress tolerance in tomato genotypes, in addition to the expression of *HSFA2, HSP70*, and *HSP90*, control of leaf cooling by transpiration and water relations, the antioxidant machinery and hormone, and secondary related signaling pathways were analyzed. Heat stress produces ROS in plant tissues [63]. In our study, an early induction of *RBOHB* was found in MO-10 plants. Interestingly, at 6 hpHS a strong induction of this gen was only observed in the thermo-tolerant genotypes (MO-10 and MM). Recent studies provide compelling evidence showing that mutation in brassinosteroid pathways impaired the induction of *Rboh1*, H_2_O_2_ apoplastic accumulation and thermotolerance, and that exogenous H_2_O_2_ fully recovered the heat stress tolerance of the mutants [64]. Although the production of ROS causes damage to cell structures, it can play an important signaling role in stress responses [65]. When ROS increase, one of the main mechanisms of thermotolerance is the upregulation of the antioxidant system [66] to generate cellular homeostasis and fix the injuries produced by HS [67]. In this work, we demonstrated that MO-10 plants displayed an early induction of *DHAR1*, which is a gene related to ascorbic acid metabolism. Recently, [67] postulated that ascorbic acid produced priming effect on tomato roots against HS by reducing the oxidative damage as well as increasing key components of thermotolerance. In addition to the accumulation of metabolites with antioxidant capacity, the upregulation of genes encoding antioxidant enzymes such as *Cu/Zn-SOD*, *Mn-SOD*, and *GR* by heat priming was effective in inducing tolerance in wheat seedlings to subsequent HS [68]. In this work, a correlation was observed between heat-tolerant genotypes and the activation of antioxidant machinery, given that a marked statistically significant increase was observed in *SODct* and *CAT* genes in MO-10 and MM genotypes, respectively, which supports the notion that control of oxidative burst produced by HS is a key point in thermotolerance. The H_2_O_2_ accumulation associated to *RBOHB* activation could be crucial for the heat-tolerant phenotype of MO-10 plants, since the accumulation of this signaling molecule could be related to the activation of HS response. However, control of the damage associated to the oxidative burst through the accumulation of antioxidant enzymes (only observed in MM and MO-10) is important to reduce oxidative stress. 

The metabolic profile of tomato plants from different genotypes against HS provide a clue to better understand the signaling networks that act in the intra-genetical variation of HS responses. The increase of endogenous ABA concentration after heat stress enhances the antioxidant ability to confer heat tolerance in plants [69]. In this work, *ASR1* expression and ABA content were evaluated to clarify the role of ABA-dependent signaling pathways on the contrasting levels of heat tolerance obtained among tomato genotypes. The heat-tolerant phenotype displayed a great *ASR1* activation at 2 and 6 hpHS, although ABA accumulation was not observed at the studied time points. However, it has been observed an *ASR1* and *RBOHB* expression profiles correlation in these genotypes. These results are consistent with those published by [70], who observed that ABA-treated plants showed a higher accumulation of H_2_O_2_ that mediated the induction of heat tolerance. For that, we speculated that the activation of ABA-dependent signaling pathways (based on the induction of *ASR1* as an ABA marker) as well as H_2_O_2_-mediated responses (based on *RBOHB* activation) is an important mechanism of thermotolerance. 

Moreover, phenolic compounds are an important class of plant secondary metabolites, which play crucial physiological roles throughout the plant life cycle. The analysis of phenylpropanoid pathway revealed that MM plants displayed an early *PAL* upregulation, which was correlated with a CGA accumulation at 4 hpHS. A recent study postulated that the accumulation of phenolic compounds in *Festuca trachyphylla* subjected to HS was accompanied by enhanced tolerance to the increase of temperature [71], and other works concluded that their implication on heat responses relies on their capacity to prevent heat induced oxidative damage [72]. According to these findings, early phenylpropanoid pathway activation in MM plants could have an important role in the thermotolerant phenotype, despite the initial damage due to HS.

To better understand the role of the activated pathways throughout HS, we also studied the mechanisms involved in the recovery phase. Although the analysis of the photosynthetic parameters in this phase confirmed the genotype variation in responses, the study of the active pathways after stress provided very interesting information about how plants recover their structures after an adverse situation. Although it is widely known the implication of SA in abiotic stress tolerance (review by [73]), little investigation of the SA-mediated responses in the recovery phase has been conducted. The widespread SA accumulation at 6+2 hpHS in all the genotypes analyzed regardless of their phenotype under HS supports that SA-related pathways might have an important role in ameliorating post-stress heat-induced cellular damage. Foliar application of SA 1 mM on *Cucumis sativa* produced a decrease on electrolyte leakage and oxidative stress, and improved maximum yield of PSII, Fv′/Fm′ and the quantum yield of the PSII electron transport after both HS and in the recovery phase [22]. In higher plants, SA is synthesized by two distinct pathways, *PAL*- and ISC-pathways [74,75]. Interestingly, the highest levels of *PAL* expression at recovery phase occurred in TH-30 and ADX2 plants with a heat-sensitive phenotype. Moreover, the high expression level of *PAL* in TH-30 and ADX2 plants correlates with a high accumulation of phenolic compounds such as CGA and FA at 6 hpHS and in the recovery phase. Although, the positive effect of phenolic compounds in the plants’ responses against heat stress is well documented [46], our findings support that these secondary metabolites are one of the keys to overcoming the damage caused by high temperatures. When we searched the mechanisms activated in the post-stress period, we found that *ASMT*, a gene related with the melatonin biosynthetic pathway, was strongly induced in all the tomato genotypes with respect to the HS period. Recent studies have demonstrated that melatonin treatment of tomato seedlings could enhance glucose metabolism, improve photosynthetic efficiency, upregulate melatonin biosynthesis, and reduce the photoinhibition occurred in the HS [76]. Therefore, our data support that the activation of this metabolic pathway is an important event for plant recovery from leaf damage produced under HS conditions. 

## 4. Materials and Methods

### 4.1. Plant Materials and Growth Conditions

Commercial cultivars *Money Marker* (MM) and *Ailsa craig* (Ailsa) were used as a reference and four traditional tomato varieties, originated from the Mediterranean area, TH-30 (Greece); ADX2 (Spain); ISR-10 (Israel); MO-10 (France) were selected for the study. Tomato varieties were obtained from the COMAV-UPV (Institute for the Preservation and Improvement of Valencian Agro-Diversity, Universidad Politécnica de Valencia).

Tomato seeds were germinated in vermiculite in a growth chamber under the following environmental conditions: light/dark cycle of 16/8 h, temperature of 26/18 °C, light intensity of 200 µmol m^−2^ s^−1^, and a relative humidity of 60%. The seeds were irrigated twice a week with distilled water. Ten days after germination and during the three next weeks, tomato genotypes were irrigated twice a week with Hoagland solution [77] applied at the same proportion.

Four-week-old tomato plants were transferred to another growth chamber to apply HS. In order to confirm that all tomato plants started from the same substrate humidity, 24 h before the HS, the pots were saturated with Hoagland solution and the excess was removed before the start of the heat shock process. To achieve HS, the temperature of the culture chamber was increased to 42 °C during 6 h and the recovery capacity was subsequently evaluated after 2 h under normal temperature conditions. The phenotype evaluation and the collection of plant material were carried out at 0, 2, 4, and 6 h post-heat shock (2, 4, and 6 hpHS), as well as two hours after the end of the stress in recovery phase (6+2 hpHS).

### 4.2. Visual Damage

Four damage levels were established depending on the leaf roll level: level 0 (healthy), level 1 (minor roll), level 2 (medium roll) and level 3 (severe roll or dead). The phenotype was determined by the percentage of leaflets of the third and fourth true leaf with symptoms in relation to the total number of analyzed leaflets.

### 4.3. Relative Water Content (RWC)

Apical leaflet from the third fully expanded leaf of five plants per cultivar was cut from a plant at 0, 2, 4, 6, and 6+2 hpHS. Fresh weight (FW) of the leaflet was immediately measured after cutting. Then, the leaflet was immersed in dd-H_2_O and incubated overnight under normal room temperature. The leaflet was taken out, properly dried to remove the water drops from the surface of the leaf and weighed to obtain the turgid weight (TW). Immediately, the leaflet was put in a drying oven for 24 h and weighed to obtain dry weight (DW). Relative water content was calculated as (RWC in %) = [(FW − DW)/(TW − DW)] × 100.

### 4.4. Photosynthetic Parameters

Determinations were carried out in situ on the apical part of leaves of the same age belonging to four-week-old tomato plants. During the gas exchange measurements, plants were maintained in the chamber where the HS was applied. The gas exchange analysis was carried out using a portable open system infrared gas analyzer (LI-6800 portable photosynthesis system, LI-COR, USA) under ambient CO_2_, light intensity (150 µmol m^−2^ s^−1^) and humidity. The parameters of interest were: The photosynthetic rate (A, μmol CO_2_ m^−2^ s^−1^), transpiration rate (E, mol H_2_O m^−2^ s^−1^), stomatal conductance (gs, mol H_2_O m^−2^ s^−1^), electron transport rate (ETR, µmol of electrons m^−2^ s^−1^), maximum quantum efficiency of photosystem II (Fv′/Fm′), and leaf temperature (Tleaf, °C). The results were obtained by taking 3 measures per leaf on three different leaves. The experiment was repeated 3 times (n = 27). 

### 4.5. Gene Expression

Gene expression by qRT-PCR was performed on the RNA samples extracted from tomato leaves using the Total Quick RNA Cells and Tissues kit (E.Z.N.A. Mini kit; http://omegabiotek.com), according to the manufacturer’s instructions. The tomato leaf samples for RNA isolation were collected at 0, 2, 4, 6, and 6+2 hpHS. Highly pure RNA was used for the RT reaction. The RT reaction was performed according to the manufacturer’s instructions for the Omniscript Reverse Transcriptase kit (QIAGEN; http://www.qiagen.com/). The primers used for the qRT-PCR are listed in Table 1. As an internal housekeeping control, the expression of EF1α gene was used. At least three independent experiments were performed to confirm the results.

### 4.6. Chromatographic Analysis 

For the hormonal analysis, fresh material was frozen in liquid N, ground, and freeze-dried. Fresh tissue (0.5 g) was immediately homogenized in 2.5 mL of ultrapure water, and 100 ng mL^−1^ of a mixture of internal standards ((2H6-ABA (to quantify ABA), 2H4-SA (to quantify SA and propylparaben (to quantify phenolic compounds like ferulic acid (FA) and chlorogenic acid (CGA)), (Sigma–Aldrich, St. Louis, MO, USA)) were added prior to extraction. The samples were centrifuged at 5000 rpm for 45 min at 4 °C. The supernatant was partitioned against diethylether, dried in a speed vacuum and resuspended in 90:10 H_2_O:MeOH. [78]. After extraction, a 20 µL aliquot was injected directly into an ultra-high performance liquid chromatography (UPLC) system with an ACQUITY UPLC BEH C18 column (1.7 μm 2.1 × 50 mm) (Waters, Mildford, MA, USA), which was interfaced with a triple quadrupole mass spectrometer (TQD, Waters, Manchester, United Kingdom). Version 4.1 of the MASSLYNX NT software (Micromass) was used to process the quantitative data from the calibration standards and plant samples. The concentrations of hormones and phenolic compounds were determined in each sample by normalizing the chromatographic area for each compound with the fresh weight of the corresponding sample. 

### 4.7. Statistical Analysis

Statistical analysis was carried out by one-way ANOVA using the Statgraphics plus software for Windows, V.5 (Statistical Graphycs Corp., Maryland, MA, USA). The means were expressed with a standard error (SE). The comparison was carried out using Fisher’s least significant difference (LSD) at 95%. Differences were taken into account only when there were significant at the 5% level. Each experiment was repeated at least three times.

## 5. Conclusions

As a result of climate change, crops are predicted to be exposed to high temperatures more frequently. Hence, it is essential to explore how tomato genotypes respond to high temperatures to select those that will be better suited for future climate change. In this way, the use of traditional varieties could be a promising approach to understand the complex response to heat stress. After the screening of the response of several tomato landraces against heat stress, we identified that their thermotolerance strategies are very different (Figure 9). MO-10 plants could maintain the photosynthetic parameters without variations during the early stages of HS, which kept the Tleaf within an optimal range for normal functions without damage on the photosystems. Moreover, we observed an early activation of H_2_O_2_ and ABA-related signaling pathways together with an induction of SOD levels to control the oxidative burst. However, in MM plants, a sudden and strong reduction of photosynthetic parameters was observed at 2 hpHS, triggering leaf temperature increase and producing severe damage on the photosystems. On the other hand, the synergic activation of *HSP9*0 and *HSP70* genes, as well as *PAL* induction probably boosted the overcoming stress, showing a recovery of E and gs parameters that participate in transpirational cooling at 4 hpHS, thus reducing leaf temperature. In this case, to control the HS-mediated oxidative burst, MM plants synthesized FA and induced the synthesis of *CAT*. Moreover, in this work it was demonstrated that SA-dependent pathways, as well as the synthesis of phenolic compounds and melatonin, are key points in the recovery processes after severe heat stress. Therefore, the characterization of traditional and commercial varieties against HS allows to find heat tolerant genotypes of tomato for use in combating the challenge of climate change. Moreover, as described above, the reproductive organs of plants are highly sensitive to heat stress, producing serious losses in crop productivity. To extrapolate the results of this work to field and to gain insight into the behavior of traditional varieties for their characterization as thermotolerant or thermosensitive, future research could be directed to elucidate the effect of heat stress on their reproductive organs to be included into breeding programs. Moreover, understanding the molecular events related to thermotolerance as well as the mechanisms responsible for cell recovery after stress are key to generating more resistant and resilient crops to extreme environmental episodes.

## Figures and Tables

**Figure 1 plants-11-03170-f001:**
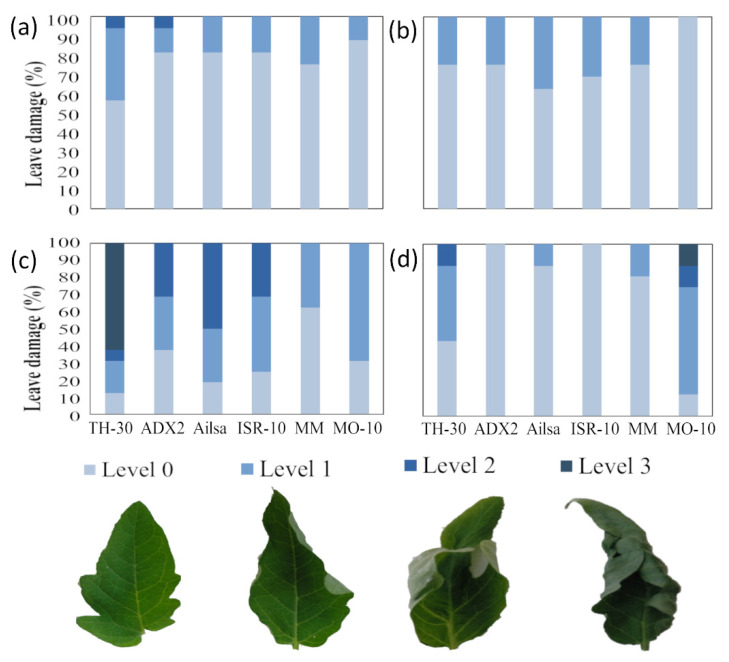
Evaluation of leaf damage provoked by HS on the six tomato genotypes. Leaf damage was measured using a four-level severity scale that identifies the damage depending on the leave roll: level 0 (healthy), level 1 (minor roll), level 2 (medium roll) and level 3 (severe roll or dead). The photograph shows a representative picture of damage level. Leaf damage produced by the increase of temperature until 42 °C were measured as a percentage of the leaflets with symptoms in relation to the total number of analyzed leaves at (**a**) 2 hpHS, (**b**) 4 hpHS, (**c**) 6 hpHS, and (**d**) 6+2 hpHS. Data show the average of three independent experiments, with values of 10 seedlings per experiment.

**Figure 2 plants-11-03170-f002:**
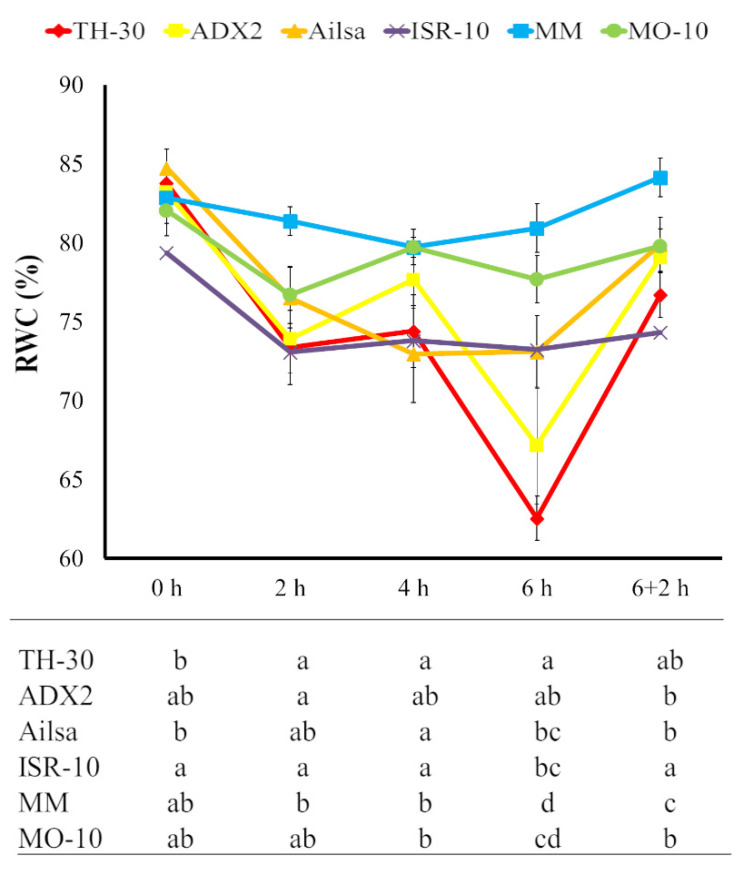
Relative water content (RWC, %) of the six tomato genotypes upon HS. HS was applied on four-week-old tomato plants by the increase of 42 °C in the culture chamber during 6 h and the recovery capacity was subsequently evaluated after 2 h under normal temperature conditions. Leaflets were collected at various time points and the RWC were evaluated. Data show the average of three independent experiments, with values of 10 seedlings per experiment ±SE. Letters in the table indicate statistically significant differences between genotypes at each time point (*p* < 0.05; LSD test).

**Figure 3 plants-11-03170-f003:**
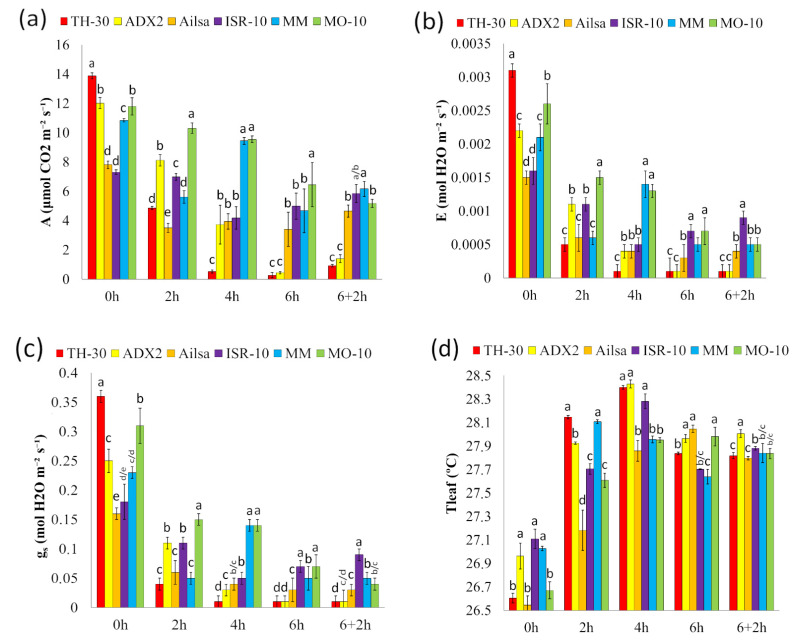
Photosynthetic parameters of the six tomato genotypes during the HS. The stress was applied to four-week-old tomato plants by increasing the culture chamber temperature to 42 °C during 6 h, and the recovery capacity was subsequently evaluated after 2 h under normal temperature conditions. The measurements of (**a**) photosynthetic rate (A: μmol CO_2_ m^−2^ s^−1^); (**b**) transpiration rate (E: mol H_2_O m^−2^ s^−1^); (**c**) stomatal conductance (g_s_: mol H_2_O m^−2^ s^−1^); leaf temperature (Tleaf: °C); were taken at different time points. Data are expressed as mean ± SE from three biological replicates, each replicate consisting of 3 plants with three technical replicates (n = 27). Letters indicate statistically significant differences between genotypes at each time point (*p* < 0.05; LSD test).

**Figure 4 plants-11-03170-f004:**
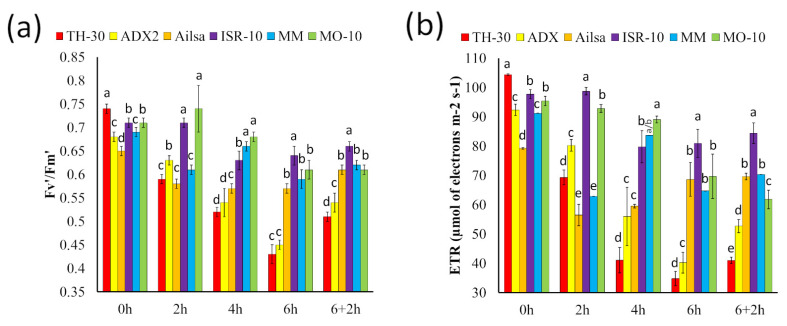
Impact of HS on the photosystems. Tomato plants were grown, and the HS was applied as described in Figure 2. The measurements of (**a**) maximum quantum efficiency of photosystem II (Fv′/Fm′) and (**b**) electron transport rate (ETR, µmol of electrons m^−2^ s^−1^) were taken at different time points. Data are expressed as mean ± SE from three biological replicates, each replicate consisting of 3 plants with three technical replicates (n = 27). Letters indicate statistically significant differences between genotypes at each time point (*p* < 0.05; LSD test).

**Figure 5 plants-11-03170-f005:**
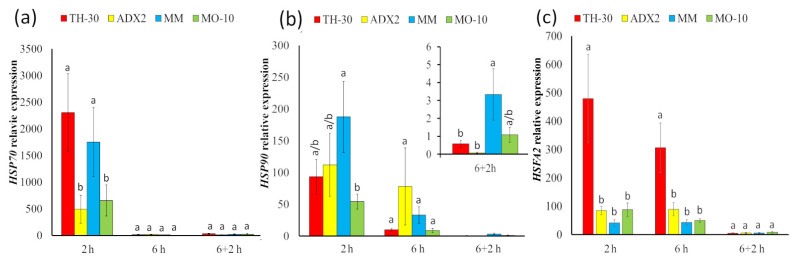
Effect of genotype on the heat responses related genes of tomato plants under HS. Tomato plants were grown, and the HS was applied as described in Figure 2. Total RNA was isolated from leaves at different time points, converted cDNA and subjected to qRT-PCR analysis. The expression levels of genes (**a**) *HSP70*, (**b**) *HSP90*, and (**c**) *HSFA2* were analyzed. The relative expression levels of each gene were normalized to those of *EF1α*. Letters indicate statistically significant differences between genotypes at each time point (*p* < 0.05; LSD test).

**Figure 6 plants-11-03170-f006:**
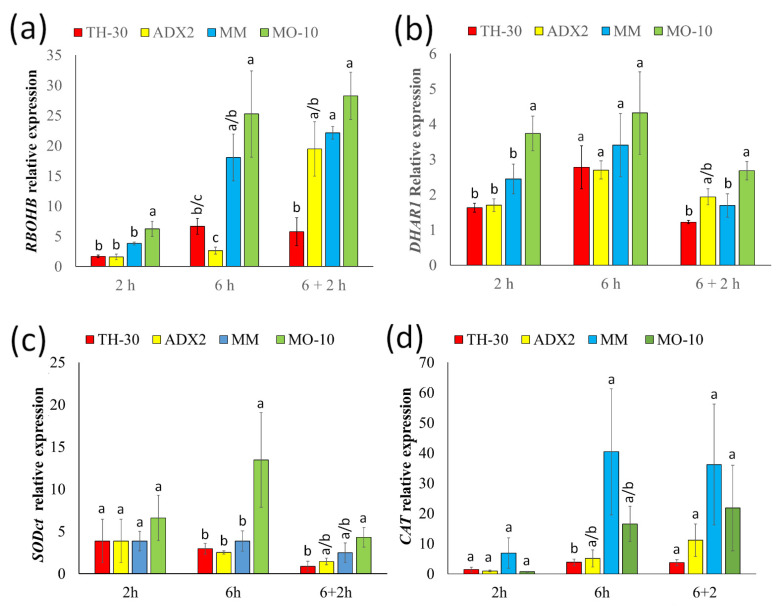
Expression profile of the genes involved in the antioxidant system in the four tomato genotypes upon/under HS. Tomato plants were grown, and the HS was applied as described in Figure 2. Total RNA was isolated from leaves at different time points, converted cDNA and subjected to qRT-PCR analysis. The expression levels of genes (**a**) *RBOHB*, (**b**) *DHAR1*, (**c**) *SODct*, and (**d**) *CAT* were analyzed. The relative expression levels of each gene were normalized to those of *EF1α.* Letters indicate statistically significant differences between genotypes at each time point (*p* < 0.05; LSD test).

**Figure 7 plants-11-03170-f007:**
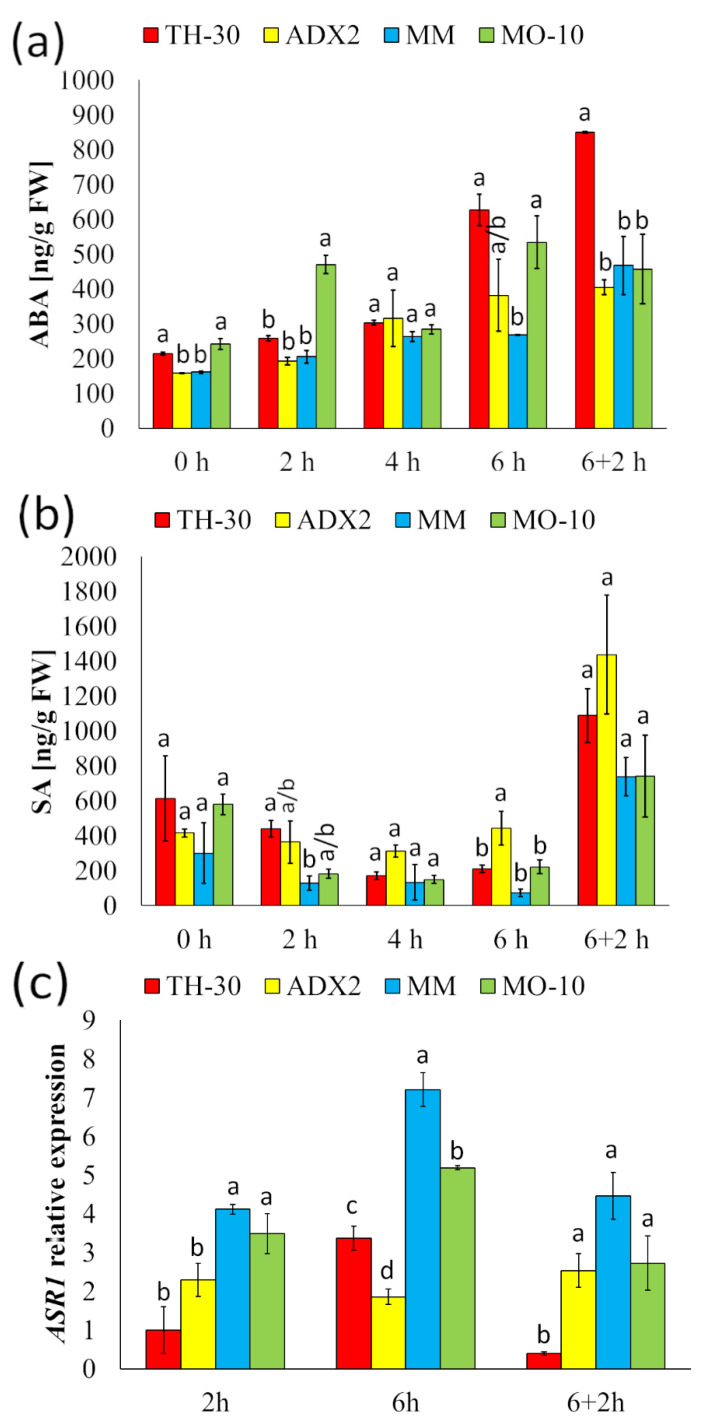
Effect of genotype on the hormonal response of tomato plants under HS. Tomato plants were grown, and the HS was applied as described in Figure 2. Leaves were collected at different time points, and (**a**) ABA and (**b**) SA levels were determined using UPLC–MS. The relative levels of (**c**) *ASR1* were analyzed and normalized to the *EF1α* gene expression level measured in the same sample. Letters indicate statistically significant differences between genotypes at each time point (*p* < 0.05; LSD test).

**Figure 8 plants-11-03170-f008:**
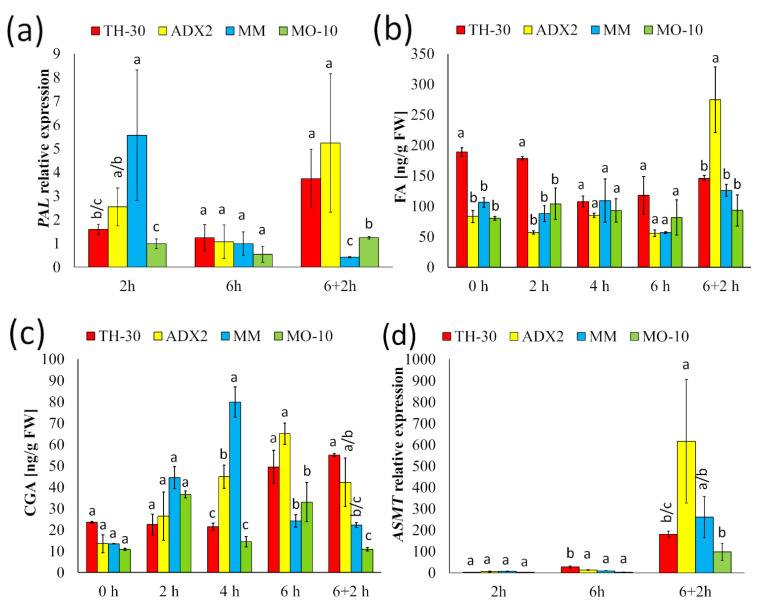
Effect of genotype on the variation of secondary metabolisms related responses of tomato plants under HS. Tomato plants were grown, and the HS was applied as described in Figure 2. Leaves were collected at different time points, and (**a**) *PAL* expression, (**b**) CFG, and (**c**) FA levels were determined. Moreover, the relative levels of (**d**) *ASMT* were analyzed. The relative expression levels of each gene were normalized to those of *EF1α.* Letters indicate statistically significant differences between genotypes at each time point (*p* < 0.05; LSD test).

**Figure 9 plants-11-03170-f009:**
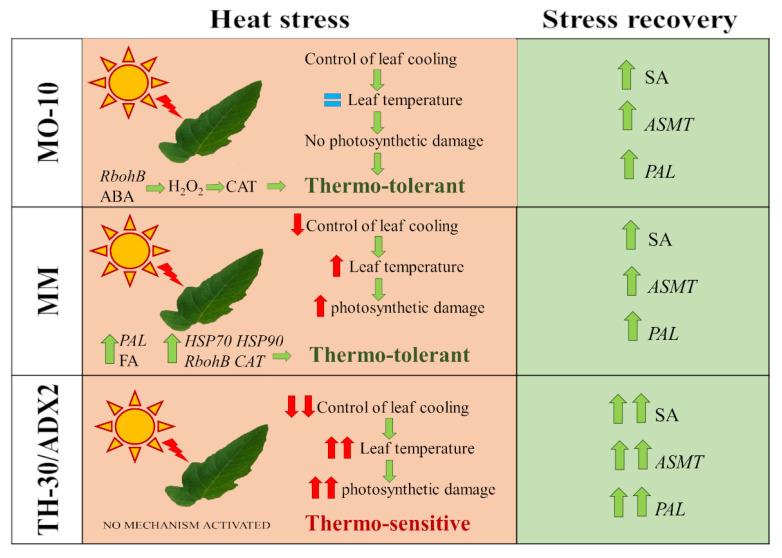
Comparative diagram between the mechanisms activated in MM and MO-10 (thermo-tolerant genotypes), and TH-30 and ADX2 (thermo-sensitive genotypes) after heat stress.

**Table 1 plants-11-03170-t001:** Primers used for plant gene expression analyses.

Function	Gene	Primer
Heat shock protein 70	*HSP70*	F 5′-GCATTGCCGGATTAGATGTT-3′ R 5′-CATCACCTCCCAAGTGTGTG-3′
Heat shock protein 90	*HSP90*	F 5′-CTTGGATTCGTGAAGGGTGT-3′ R 5′-GCCCAGCTTCAAGTTCTTTG-3′
Respiratory burst oxidase B	*RBOHB*	F 5′-AGGGAATGATAGAGCGTCG-3′ R 5′-CATCGTCATTGGACTTGGC-3′
Dehydroascorbate reductase 1	*DHAR1*	F 5′-AGGTGGCTCTTGGACACTTC-3′ R 5′-CTTCAGCCTTGGTTTTCTGG-3′
Heat shock transcription factor A2	*HSFA2*	F 5′-GATCTGGTGCTTGCATTGAA-3′ R 5′-TGGGGGTCATCGTTAGTCTC-3′
Catalase	*CAT*	F 5′-TGCATTGAAACCAAATCCAA-3′ R 5′-TGTGCTTTCCCCTCTTTGTT-3′
Superoxide dismutase 1	*SODct*	F 5′-GGAAAGGGAGGACATGAGCT-3′ R 5′-ACCCCAATTCAAAAGGCGTC-3′
Phenylalanine ammonia-lyase 1	*PAL*	F 5′-CAAGAATTAGATGCCTTAACCAA-3 R 5′-ACTATTCAAAAGGTCCATCAGTTT-3′
Acetylserotonin O-methyltransferase	*ASMT*	F 5′-GCATGGCTGCACTTGTCTTA-3′ R 5′-ATGCTCCGGATTGATTTTTG-3′
Abscisic stress-ripening	*ASR1*	F 5′-ACACCACCACCACCTGT-3′ R 5′-GTGTTTGTGTGCATGTTGTGGA-3′
Elongation factor 1-alpha	*EF1α*	F 5′-GACAGGCGTTCAGGTAAGGA-3′ F 5′-GGGTATTCAGCAAAGGTCTC-3

## Data Availability

Not applicable.

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
