# Peer review of "Exploiting Tomato Genotypes to Understand Heat Stress Tolerance"

_plants, 2022, doi:10.3390/plants11223170_

Round 1

Reviewer 1 Report

This article aimed to elucidate heat stress tolerance mechanisms of tomato. Firstly, the authors classified 6 tomato genotypes into heat-sensitive, intermediate, and heat -tolerant based on visual symptoms, RWC, leaf cooling, and photosynthetic parameters. Molecular studies were then conducted on two heat-sensitive and two heat-tolerant genotypes to elucidate heat tolerant mechanisms in the two tolerant genotypes. The experimental design and methodologies were sound. Generally, the data were accurately collected and interpreted. Expression of genes related to several mechanisms/pathways were studied, as well as related biochemical analysis. The results mostly supported the conclusion that the tolerant genotype MO-10 primarily employed leaf cooling mechanism together with the early activation of ABA signaling pathway, while the MM variety relied on the activation of heat shock proteins and efficient ROS scavenging based on secondary metabolites like phenolics and antioxidant enzymes.

               In my opinion, this paper presented an interesting set of data to help elucidate how the tolerant tomato tolerated heat during stress and suggested how the plants recovered after stress. This article should attract attentions from a wide audience, both physiologists and molecular biologist. The paper is of good enough quality to be accepted for publication in Plants, after some minor revisions as follows:

1.      Method section – the parameter Fv/Fm has to be measured after the leaves were kept in dark (dark adaptation) for 20-30 minutes. The authors did not mention when the dark adaptation was performed (before or after leaf gas exchange measurement?). In relation to this point, in the Result section, the text described Figure 4(a) as Fv/Fm, but the Y axis in this Fig.4(a) is labelled as Fv’/Fm’ (the quantum efficiency of PSII measured in the light). This created a confusion, which parameter was actually measured? In fact, the value Fv/Fm before stress was abnormally low (it should be very close to 0.8 for healthy plants), and it is surprising that this value decreased so sharply and so abruptly within hours of stress.

2.      Method section – the light intensity during gas exchange measurement was not indicated.

3.      Method section – it would be useful for discussion if the author also mentioned the level of damage (level 0, 1, 2 or 3) of the leaves collected for the physiological/molecular determination.

4.      Method section – primers for asr1 gene were missing.

5.      Figure 4(a) – the Y axis was labelled as Fv’/Fm’ as mentioned above

6.      Figure 5 – the graph for RBOHB expression should be removed

7.      Figure 3(d) – the label for var. ISR-10 was shown as ‘israel’, and MO-10 as Mo

8.      Figure 7 – graphs SA level and asr1 expression should be re-ordered.

9.      A lot of inconsistencies were apparent for many technical terms, for example,

RBOHB VS RbohB; quantum efficiency of photosystem II (Fv/Fm) VS maximum quantum efficiency of Photosystem II (Fv/Fm); leaf To VS Tleaf; gene names were not italicized in many places including on Y axis in the figures; any reason why the asr1 gene was typed in lowercase letter, not ASR1?; Fv/Fm was mistyped as FV/Fm in many places etc.

10.   Discussion part: please rewrite the part concerning the relationship between ABA level and expression of asr1 gene for clearer understanding

11.   Abstract – the sentence “The MO-10 plants tolerance was based on the control of the leaf cooling mechanism and the rapid activation of H2O2 and ABA signalling pathways.” should be modified. The authors did not directly measure H2O2. It is not the activation of H2O2 but activation of RBOH leading to H2O2 accumulation in turn acting as a signal molecule.

12.   The language should be improved, and typing errors should be attended to.

Author Response

Comments and Suggestions for Authors REVIWE 1

This article aimed to elucidate heat stress tolerance mechanisms of tomato. Firstly, the authors classified 6 tomato genotypes into heat-sensitive, intermediate, and heat -tolerant based on visual symptoms, RWC, leaf cooling, and photosynthetic parameters. Molecular studies were then conducted on two heat-sensitive and two heat-tolerant genotypes to elucidate heat tolerant mechanisms in the two tolerant genotypes. The experimental design and methodologies were sound. Generally, the data were accurately collected and interpreted. Expression of genes related to several mechanisms/pathways were studied, as well as related biochemical analysis. The results mostly supported the conclusion that the tolerant genotype MO-10 primarily employed leaf cooling mechanism together with the early activation of ABA signaling pathway, while the MM variety relied on the activation of heat shock proteins and efficient ROS scavenging based on secondary metabolites like phenolics and antioxidant enzymes.

               In my opinion, this paper presented an interesting set of data to help elucidate how the tolerant tomato tolerated heat during stress and suggested how the plants recovered after stress. This article should attract attentions from a wide audience, both physiologists and molecular biologist. The paper is of good enough quality to be accepted for publication in Plants, after some minor revisions as follows:

  1. Method section – the parameter Fv/Fm has to be measured after the leaves were kept in dark (dark adaptation) for 20-30 minutes. The authors did not mention when the dark adaptation was performed (before or after leaf gas exchange measurement?). In relation to this point, in the Result section, the text described Figure 4(a) as Fv/Fm, but the Y axis in this Fig.4(a) is labelled as Fv’/Fm’ (the quantum efficiency of PSII measured in th e light). This created a confusion, which parameter was actually measured?

The parameter measured in this work is the Fv'/Fm' (the quantum efficiency of PSII measured in the light). It was a typographical error and it has been modified throughout the text

In fact, the value Fv/Fm before stress was abnormally low (it should be very close to 0.8 for healthy plants), and it is surprising that this value decreased so sharply and so abruptly within hours of stress.

Although the usual values of this parameter in healthy plants are around 0.8, Olvera-González et al., 2013 showed that the values of Fv'/Fm' in tomato plants in the absence of stress were between 0.65 and 0.70.

Chlorophyll fluorescence emission of tomato plants as a response to pulsed light based LEDs. Plant Growth Regul 69, 117–123 (2013). https://doi.org/10.1007/s10725-012-9753-8

  1. Method section – the light intensity during gas exchange measurement was not indicated.

The light intensity has been added to this sentence of material and methods:

The gas exchange analysis was carried out using a portable open system infrared gas analyser (LI-6800 portable photosynthesis system, LI-COR, USA) under ambient CO2, light intensity (150 µmol m‑2 s‑1) and humidity

  1. Method section – it would be useful for discussion if the author also mentioned the level of damage (level 0, 1, 2 or 3) of the leaves collected for the physiological/molecular determination.

A sentence from the discussion has been rewritten to clarify this point:

Specifically, the visual evaluation of foliar damage revealed that the most affected genotypes by the increase in temperature were TH-30 and ADX2, showing 100% and 70% of its leaflets damaged in level 3 respectively, and the most tolerant were MM and MO-10, since these genotypes showed less than 60% damaged leaflets observing only 20% of them in levels 2 and 3 respectively

  1. Method section – primers for asr1gene were missing.

ASR1 primers have been added to table 1.

  1. Figure 4(a) – the Y axis was labelled as Fv’/Fm’ as mentioned above

As described above, this is the correct nomenclature for the measured parameter.

  1. Figure 5 – the graph for RBOHB expression should be removed.

The graph of RBOHB has been removed from Figure 5

  1. Figure 3(d) – the label for var. ISR-10 was shown as ‘israel’, and MO-10 as Mo

The legend has been modified

  1. Figure 7 – graphs SA level and asr1 expression should be re-ordered.

The figures have been re-ordered

  1. A lot of inconsistencies were apparent for many technical terms, for example,

RBOHB VS RbohB; quantum efficiency of photosystem II (Fv/Fm) VS maximum quantum efficiency of Photosystem II (Fv/Fm); leaf To VS Tleaf; gene names were not italicized in many places including on Y axis in the figures; any reason why the asr1 gene was typed in lowercase letter, not ASR1?; Fv/Fm was mistyped as FV/Fm in many places etc.

Gene names and other typographical errors have been corrected throughout the text. In addition, the axes of graphs 3, 5, 6 and 7 have been modified.

  1. Discussion part: please rewrite the part concerning the relationship between ABA level and expression of asr1 gene for clearer understanding.

The discussion sentence has been rewritten and replaced with the following:

Heat-tolerant phenotype displayed a great ASR1 activation at 2 and 6 hpHS, although ABA accumulation is not observed at the studied time points. However, it has been observed an ASR1 and RBOHB expression profiles correlation in these genotypes. These results are consistent with those published by [71], who observed that ABA-treated plants showed a higher accumulation of H2O2 that mediated the induction of heat tolerance. For that, we speculated that the activation of ABA-dependent signalling pathways (based on the induction of ASR1 as an ABA marker) as well as H2O2-mediated responses (based on RBOHB activation) is an important mechanism of thermotolerance.   

  1. Abstract – the sentence “The MO-10 plants tolerance was based on the control of the leaf cooling mechanism and the rapid activation of H2O2and ABA signalling pathways.” should be modified. The authors did not directly measure H2O2. It is not the activation of H2O2 but activation of RBOH leading to H2O2 accumulation in turn acting as a signal molecule.

The sentence of the abstract has been modified

  1. The language should be improved, and typing errors should be attended to.

The entire manuscript has been revised to correct these issues.

Reviewer 2 Report

The authors conducted essential and current research.
The processing of the topic is logical and mostly followable. The conclusions are sympathetically
 moderate. The technical approach is simple but presumably effective. Of course, the question arises as to why we think transcriptional responses are "exclusive" under stress conditions. It should be evident that post-translational modifications of proteins are several orders of magnitude more cost-effective than this. Of course, it would be unexpected to examine them in this publication.
Figure 5 contains figure 6a. Not all figures are well labelled.

Author Response

The authors conducted essential and current research.
The processing of the topic is logical and mostly followable. The conclusions are sympathetically
 moderate. The technical approach is simple but presumably effective. Of course, the question arises as to why we think transcriptional responses are "exclusive" under stress conditions. It should be evident that post-translational modifications of proteins are several orders of magnitude more cost-effective than this. Of course, it would be unexpected to examine them in this publication.

We agree with the comment made by the reviewer, since almost certainly the condition of thermos-tolerant or thermos-sensitive cultivar may be determined by other parameters such as post-translational protein modifications. It would be really interesting to analyze these mechanisms and how they influence the acquisition of thermotolerance.

Figure 5 contains figure 6a. Not all figures are well labelled.

Figures have been revised and modified for publication.

Reviewer 3 Report

In the Ms, Exploiting tomato genotypes to understand heat stress tolerance, authors have analysed the heat-sensitive: TH-30, ADX2; intermediate: ISR10 and Ailsa Craig; heat-tolerant: MM and MO-10 tomato varieties at various scale.

Overall, it is good study but not new. These kind of several studies have been published earlier.

Authors should compare these varieties with earlier known varieties for different features.

And the most important part is yield, which has not been shown here. Authors should first see the yield, the remaining analysis are secondary if yield is not upto the mark.

Author Response

In the Ms, Exploiting tomato genotypes to understand heat stress tolerance, authors have analysed the heat-sensitive: TH-30, ADX2; intermediate: ISR10 and Ailsa Craig; heat-tolerant: MM and MO-10 tomato varieties at various scale.

Overall, it is good study but not new. These kind of several studies have been published earlier.

Authors should compare these varieties with earlier known varieties for different features.

And the most important part is yield, which has not been shown here. Authors should first see the yield, the remaining analysis are secondary if yield is not upto the mark.

Despite not being the first study carried out on traditional tomato varieties, we believe it is interesting to note that the importance of the study lies in the analysis of the behaviour of these varieties from the Mediterranean basin, since it is known that this area will be strongly affected by the climate change.

In this work we pretend to study the response mechanism of plants with different phenotypes to heat stress and, although it is interesting to see the production, the experiments are carried out in a culture chamber to be able to correctly control the test conditions, so that they are homogeneous (temperature, humidity, duration of the ramps in temperature changes). We focusing our investigation in the molecular mechanisms related to this thermotolerance, for this reason we have carried out the analyses on a one-month-old plant. However, It would be very interesting to analyse in future research the effect of the increase in temperature on production, in experiments carried out under controlled conditions in the greenhouse.

Round 2

Reviewer 3 Report

The yield component is very important when we study crop plants, especially for stress tolerance. If authors have analyzed one-month-old plants, they can wait for a few more months for yield analysis, why they are in hurry to publish? It should be included in this Ms. This is my view, the editor may have a different view of your study. 

Author Response

It is true that the reproductive apparatus of plants is seriously affected by the increase in high temperatures and that this causes losses in production. We agree with the reviewer's comments. Therefore, we plan to address this topic in future research, in order to point out these varieties as very interesting in tomato breeding programs. In addition, we focus on the analysis in young plants at short times and in the recovery phase, to delve into the molecular response of plants to high temperature stress and expand knowledge in the understanding of thermotolerance mechanisms. To highlight the importance of productivity in the search for thermotolerant varieties, two paragraphs have been included in the text marked in purple.
